# Differential Role of NKG2A/HLA-E Interaction in the Outcomes of Bladder Cancer Patients Treated with *M. bovis* BCG or Other Therapies

**DOI:** 10.3390/biomedicines13010156

**Published:** 2025-01-10

**Authors:** Inmaculada Ruiz-Lorente, Lourdes Gimeno, Alicia López-Abad, Pedro López Cubillana, Tomás Fernández Aparicio, Lucas Jesús Asensio Egea, Juan Moreno Avilés, Gloria Doñate Iñiguez, Pablo Luis Guzmán Martínez-Valls, Gerardo Server, Belén Ferri, José Antonio Campillo, María Victoria Martínez-Sánchez, Alfredo Minguela

**Affiliations:** 1Immunology Service, Clinical University Hospital Virgen de la Arrixaca (HCUVA), Biomedical Research Institute of Murcia (IMIB), 30120 Murcia, Spain; irl_98@hotmail.com (I.R.-L.); lgarias@um.es (L.G.); josea.campillo@carm.es (J.A.C.); vickyms7@hotmail.com (M.V.M.-S.); 2Human Anatomy Department, Universidad de Murcia and Campus Mare Nostrum, 30071 Murcia, Spain; 3Urology Service, Virgen de la Arrixaca University Clinical Hospital (HCUVA), Biomedical Research Institute of Murcia (IMIB), 30120 Murcia, Spain; alicialopezabad@gmail.com (A.L.-A.); pedrolopezcubillana@gmail.com (P.L.C.); gerardoserver@gmail.com (G.S.); 4Urology Service, Morales Meseguer Hospital, 30008 Murcia, Spain; tomas.fernandez3@carm.es; 5Urology Service, De la Vega Lorenzo Guirao Hospital, 30530 Murcia, Spain; lucasasensioegea@gmail.com; 6Urology Service, Santa Lucia Hospital, 30202 Murcia, Spain; juan.moreno2@carm.es; 7Urology Service, Los Arcos Hospital, 30739 Murcia, Spain; gloria.donate@carm.es; 8Urology Service, Reina Sofía Hospital, 30003 Murcia, Spain; pablol.guzman@carm.es; 9Pathology Service, Clinical University Hospital Virgen de la Arrixaca (HCUVA), Biomedical Research Institute of Murcia (IMIB), 30120 Murcia, Spain; belenferri@msn.com

**Keywords:** bladder cancer, immunotherapy, BCG, NK cells, NKG2A, HLA-E, patient outcomes

## Abstract

**Background**: Immunotherapy is gaining great relevance in both non-muscle-invasive bladder cancer (NMIBC), with the use of bacille Calmette–Guerin (BCG), and in muscle-invasive BC (MIBC) with anti-checkpoint therapies blocking PD-1/PD-L1, CTLA-4/CD80-CD86, and, more recently, NKG2A/HLA-E interactions. Biomarkers are necessary to optimize the use of these therapies. **Methods**: We evaluated killer-cell immunoglobulin-like receptors (KIRs) and HLA-I genotyping and the expression of NK cell receptors in circulating T and NK lymphocytes at diagnosis in 325 consecutive BC patients (151 treated with BCG and 174 treated with other therapies), as well as in 648 patients with other cancers and 973 healthy donors as controls. The proliferation and production of cytokines and cytotoxicity were evaluated in peripheral blood mononuclear cells, stimulated in vitro with anti-CD3/CD28 or BCG, from selected patients based on HLA-B −21M/T dimorphism (NKG2A ligands). **Results**: The HLA-B −21M/T genotype showed opposing results in BC patients treated with BCG or other therapies. The MM genotype, compared to MT and TT, was associated with a longer 75th-percentile overall survival (not reached vs. 68.0 ± 13.7 and 52.0 ± 8.3 months, *p* = 0.034) in BCG, but a shorter (8.0 ± 2.4 vs. 21.0 ± 3.4 and 19.0 ± 4.9 months, *p* = 0.131) survival in other treatments. The HLA-B −21M/T genotype was an independent predictive parameter of the progression-free survival (HR = 2.08, *p* = 0.01) and the OS (HR = 2.059, *p* = 0.039) of BC patients treated with BCG, together with age and tumor histopathologic characteristics. The MM genotype was associated with higher counts of circulating CD56^bright^, fewer KIR2DL1/L2^+^ NK cells, and lower NKG2A expression, but not with differential in vitro NK cell functionality. **Conclusions**: The HLA-B −21M/T is independently associated with BC patient outcomes and can help to optimize the use of new immunotherapies in these patients.

## 1. Introduction

Bladder cancers (BCs) are highly immunogenic and diverse, with 70–75% of cases presenting with recurrent non-muscle-invasive BC (NMIBC) and 25–30% with aggressive/advanced muscle-invasive BC (MIBC) [1,2,3]. Although little is still known about how immune cells respond to BC, immunotherapy with *bacille Calmette–Guerin* (BCG) remains the gold standard for carcinoma in situ (CIS) and high-risk NMIBC. Immunotherapy, particularly the use of immune checkpoint inhibitors (ICIs), has emerged as a revolutionary approach in cancer treatment, enhancing the body’s natural immune vigilance against tumor cells. ICIs work by blocking inhibitory receptors such as PD-1, CTLA-4, NKG2A, and TIGIT, which are used by tumors to evade immune surveillance, thereby promoting more effective and durable anti-tumor responses [4,5]. In the context of MIBC [6], the use of immunotherapy to target PD-1 and CTLA-4 checkpoints has shown promising results in counteracting tumor progression [6,7,8]. However, the efficacy of ICIs depends on factors such as PD-L1 expression, tumor genomic variability, the microbiome, and mutations in key signaling pathways like the JAK/STAT pathway [9]. As ICIs have demonstrated significant success in cancers such as melanoma, lung, and renal cancer, their role in BC, particularly in overcoming immune evasion mechanisms in MIBC, is gaining increasing attention [10,11], especially in combination with the blockade of the NKG2A pathway [12].

Although BCG is a potent enhancer of the immune response, only 50 to 70% of patients respond to this treatment. Therefore, understanding immunological mechanisms could contribute to improving the results [13]. Intravesical BCG triggers cellular immune response, leading to the infiltration of granulocytes, macrophages, NK cells, dendritic cells, and lymphocytes into the tumor. Once there, they secrete IL-1, IL-2, IL-6, IL-8, IL-10 IL-12, IFN-γ, TNF-α, GMCSF, and sICAM-1, propagating inflammation and inducing the apoptosis of BC cells. Dendritic cells present tumor antigens to T lymphocytes, triggering their activation and anti-tumor responses. Additionally, BCG stimulates cancer cell killing through cytotoxic T and NK cells [3]. Nonetheless, BCG treatment is associated with diverse immune-escape mechanisms [6], including the loss of MHC-I [14] or the up-regulation of PD-L1 and poliovirus receptor (PVR or CD155) in tumor cells [15,16]. The expression of CD155 is associated with a poor prognosis and enhanced tumor progression in BC [17,18]. The specific blockade of the CD155 interaction with multiple inhibitory receptors expressed on NK and T lymphocytes, such as TIGIT, killer-cell immunoglobulin-like receptor (KIR) KIR2DL5 and CD96 [19], is now being explored in various types of cancer [20].

In vitro and experimental models suggest that NK cells play important roles in BCG therapy [21,22,23,24,25,26]. Effector mechanisms of NK cells are regulated by a balance of activating and inhibitory receptors interacting with their ligands on cancer cells [27]. Depending on the KIR genotype, each individual may express varying numbers of 9 inhibitory (KIR2DL1-4, KIR2DL5a/b and KIR3DL1-3) or 6 activating (KIR2DL1-5 and KIR3DS1) receptors. Licensing interactions of inhibitory KIR and NKG2A receptors with their HLA class-I (HLA-I) ligands promote education and the full competence of NK cells [27,28,29,30,31]. NK cell licensing is associated with the (1) increased expression of CD226 [32,33]; (2) enhanced glycolysis [34]; and (3) lysosomal remodeling [35]. Licensing allows NK cells to discriminate healthy tissues from tissues expressing damage/danger signs, the loss of HLA-I (“missing-self”) [36,37], or alterations in the peptidome presented by HLA-I (“altered-self”) [38,39].

The best-characterized licensing interactions are KIR2DL1/HLA-C2 (Lys80), KIR2DL2-3/HLA-C1 (Asn80), KIR3DL1/Bw4, KIR3DL2/HLA-A3, A11 alleles, KIR2DL4/HLA-G [27], and NKG2A/HLA-E [31]. HLA-E specifically binds nonapeptides from the leader sequence of HLA-I (residues −22 to −14) with a dimorphism at position −21 [40,41]. Methionine −21 (−21M), present in all HLA-A and HLA-C and in a minority of HLA-B allotypes, provides a good anchor residue that facilitates the folding and cell-surface expression of HLA-E [42]. In contrast, threonine −21 (−21T), present in the majority of HLA-B allotypes, does not effectively bind to HLA-E. The genetic analysis of human populations worldwide shows how haplotypes with −21M rarely encode for the Bw4 or C2 ligands [31,43]. Therefore, there are two fundamental HLA haplotypes: one preferentially supplying NKG2A ligands and the other supplying KIR ligands [31]. Functional assays have shown that individuals with −21M haplotypes have NKG2A^+^ NK cells which are better educated, more phenotypically diverse, and functionally more potent than those of TT individuals [31]. HLA-B −21M/T dimorphism is associated with susceptibility to HIV [44], the killing of HIV-infected cells by NK cells [45], NK cell anti-leukemic activity, and the overall survival of acute myeloid leukemia patients under IL-2 immunotherapy [46].

However, in BC, elevated tumor expression of HLA-E is associated with increased disease progression [12,47,48], revealing the inhibitory role of the NKG2A/HLA-E interaction. This has led to this interaction being considered as a new checkpoint to be blocked with immunotherapy, and clinical trials are already ongoing. The present study explores the role that NKG2A/HLA-E interaction may have in the antitumor response in BC treated with BCG or other therapies and its usefulness for optimizing personalized immunotherapies.

## 2. Materials and Methods

### 2.1. Samples and Study Groups

This prospective observational study included 325 consecutive patients diagnosed with BC. As control groups, 925 healthy Caucasian (HC) patients and 973 patients with other types of tumor (308 melanomas, 150 myelomas, 102 pediatric acute leukemias, and 88 ovarian cancers) were included for the purpose of identifying immunological parameters that specifically influence susceptibility to BC and responses to different therapies, including BCG immunotherapy. BC tumors were classified according to the WHO Classification of Tumours of the Urinary System and Male Genital Organs [49] into (1) noninvasive urothelial neoplasms (NIUN), including CIS as well as the low- and high-grade papillary carcinomas (Ta); and (2) infiltrating urothelial carcinoma (IUC), including the NMIBC T1 stage, and the MIBC T2, T3 and T4 stages. Progression was defined in NIUN as local recurrence with a higher grade or stage, and in IUC it was defined as local recurrence with higher stage and/or development of metastatic disease. Treatment and management were conducted at the discretion of the urologists, based on the patient’s condition and tumor histology. In this study, the recurrence, progression-free survival (PFS), and overall survival (OS) were compared between patients treated with BCG and those treated with other therapies. The overall survival of patients, in addition to disease recurrence and progression, was included as an endpoint in both treatments, since the aim of our study was to evaluate the role of the NKG2A/HLA-E interaction in local and systemic tumor immune responses.

This study was approved by the Research Ethics Committee and the Institutional Review Board (IRB-00005712). Written informed consent was obtained from all patients and controls in accordance with the Declaration of Helsinki.

Peripheral blood samples anticoagulated with EDTA (for HLA and KIR genotyping and the expression analysis of NK cell receptor by flow cytometry) were obtained at diagnosis. Fresh sodium heparin blood samples were obtained from selected donors for proliferation, cytotoxicity, and cytokine production assays.

### 2.2. HLA-A, -B and -C and KIR Genotyping

HLA-A, -B, -C, and KIR genotyping were performed on DNA samples extracted from peripheral blood using the QIAmp DNA Blood Mini kit (QIAgen, GmbH, Hilden, Germany) and the Lifecodes HLA-SSO and KIR-SSO typing kits (Immucor Transplant Diagnostic, Inc. Stamford, CT, USA), as previously described [33,50]. HLA-A and HLA-B genotyping allowed us to identify alleles bearing the Bw4 motif according to the amino-acid sequences at positions 77–83 in the alpha1 domain of the HLA class-I heavy chain. Bw4 alleles with threonine at amino acid 80 (Thr80) and a higher affinity for KIR3DL1 (HLA-B*05, B*13, B*44) were differentiated from those with isoleucine 80 (Ile80) and a lower affinity (HLA-A*23, A*24, A*25, A*32 and HLA-B*17, B*27, B*37, B*38, B*47, B*49, B*51, B*52, B*53, B*57, B*58, B*59, B*63, and B*77). HLA-C genotyping allowed the distinction between HLA-C alleles with asparagine 80 (C1-epitope: HLA-C*01, 03, 07, 08, 12, 14, 16:01) and those with lysine 80 (C2-epitope: HLA- C*02, *04, *05, *06, *15, *16:02, *17, *18). Nonetheless, the KIR ligand calculator at https://www.ebi.ac.uk/ipd/imgt/hla/matching/ (accessed on 10 December 2024) was used to ascertain Bw4, C1, and C2 epitopes. Dimorphism at position −21 of the leader sequence of HLA-B was also assessed to distinguish allotypes with methionine (−21M, HLA-B*07, B*08, B*14, B*38, B*39, B*42, B*48, B*67, B*73 and B*81) from those with threonine (−21T, rest of HLA-B alleles).

KIR genotyping identified inhibitory KIRs (2DL1-L3/2DL5 and 3DL1-L3) and activating KIRs (2DS1-S5 and 3DS1), as well as KIR2DL4, which has both inhibitory and activating potential [51].

### 2.3. Expression of NK Cell Receptors in Peripheral Blood Lymphocytes

The expression of CD16, CD226 (DNAM1), NKG2A, TIGIT and KIRs (KIR2DL1, 2DS1, 2DL2/S2, 2DL3 and 3DL1) on CD56^bright^ and CD56^dim^ NK cells, as well as on CD3^+^CD4^+^ and CD3^+^CD8^+^ T cells, was analyzed simultaneously using a 12-fluorescence FACSLyric flow cytometer, LSR-II, (BD, San José, CA, USA) and DIVA^TM^ software 9.0 (BD, San José, CA, USA), following the method previously described [33,52]. The following monoclonal antibodies were used for peripheral blood labeling: CD158a-FITC (143211, R&D Systems Inc, which recognized KIR2DL1), CD158a/h-PC7 (EB6B, Beckman Coulter, which recognized both KIR2DL1 and 2DS1), CD158b2 (180701, R&D Systems Inc, KIR2DL3), CD226-PE (11A8, Biolegend), CD158e1 (DX9, R&D systems, KIR3DL1), CD16-AlexaFluor700 (3G8, BD), CD8-APC-Cy7 (SK1, BD), TIGIT-BV421 (741182, BD), CD3-BV510 (UCHT1, BD), CD4-BV605 (RPA-T4, BD), CD56 BV711 (NCAM16.2, BD), and CD159a-BV786 (131411, BD). They were incubated for 10 min at room temperature in the dark. Subsequently red cells were lysed, washed, and acquired.

### 2.4. In Vitro Functional Assays

To evaluate the impact that the HLA-B −21M/T ligand interaction had on the functionality of T and NK cells, such as the capacity to proliferate, to kill tumor cell lines (K562, J82 y T24) and to secrete cytokines, immunoefector functionality was evaluated on peripheral blood mononuclear cells (PBMCs) isolated in Ficoll density gradients from sodium heparin-anticoagulated samples of 18 donors (6 with MM, 3 with MT and 9 with TT genotype). PBMCs were stained with 0.05 µM carboxyfluorescein succinimidyl ester (CFSE, Thermo Fisher Scientific, Waltham, MA, USA) and stimulated in vitro with ImmunoCult™ Human CD3/CD28 T cell activator (StemCell Technologies, Vancouver, Canada) following the manufacturer’s instructions, or with BCG (Danish 1331, AJVaccines, Copenhagen, Denmark) at a proportion of 1:1 colony forming units (CFUs) to PBMCs. CFSE-labeled cells (1 × 10^6^/well) were incubated at 37 °C in a 5% CO_2_ incubator in 24-well flat-bottomed plates in quintuplicate. After 72 h, 1 well per sample was collected and the supernatant was stored at −80 °C and used for cytokine analysis in Luminex. After 144 h, the harvested cells were either used for cytotoxic assays or stained to analyze proliferation in a Northern Lights (NL) flow cytometer (Cytek, Amsterdam, The Netherlands).

Cytokine production in culture supernatant was analyzed using a ProcartaPlex Human Immune Monitoring 12 Plex Panel (IL-1β, IL-2, IL-4, IL-5, IL-6, IL-8, IL-10, IL-17A, IL-22, IL-23, IFN-γ, TNF-α and TGF-β1, Thermo Fisher Scientific, Vienna, Austria), following the manufacturer’s instructions, in a Luminex 300 (R&D Systems, Minneapolis, MN, USA). The analysis was performed using the Invitrogen^TM^ ProcartaPlex^TM^ Analysis App software, (Thermo Fisher Scientific, Waltham, MA, USA).

Cell proliferation was evaluated in CFSE-labeled cells after 6 days of in vitro expansion by labeling with TIGIT-BV421 (RUO, BD), CD16-V450 (3G8, BD), CD4-cFV505 (DK3, Cytek, Fremont, CA, USA), CD226-BV605 (11A8, Biolegend, San Diego, CA, USA), CD8-BV570 (RPA-T8, Biolegend), TIM-3-BV711 (7D3, BD), TCRgd-BV750 (11F2, BD), NKG2A-BV786 (131411, BD), HLA-DR-cFB548 (L2D3, Cytek), NKG2C-PE (REA205, Miltenyi Biotec, Bergisch Gladbach, Germany), CD25-cFBYG610 (BC96, Cytek), CD158bj-PE-Cy5 (GL183, Beckman Coulter, Brea, CA, USA), KIR3DL1-APC (DX9, R&D Systems Inc., Minneapolis, MN, USA), CD57-cFR668 (HNK1, Cytek), CD38-cFR685 (HIT2, Cytek), CD3-AF700 (UCHT1, BD), NKG2D-APC-H7 (1D11, Biolegend), and CD45-cFR840 (HI30, Cytek) monoclonal antibodies during 15 min at room temperature. Cells were washed with FACsFlow (BD) and acquired for analysis in an NL-flow cytometer (Cytek) and then analyzed with FACSDiva^TM^ software (BD). Proliferation was estimated as the percentage of CFSE-low cells within each cell subset (CD4+ and CD8+ T cells, CD56^dim^ and CD56^bright^ NK cells and NoT-NoNK cells) as shown in Figure 1A.

The cytotoxic activity of harvested cells (effectors) was evaluated against target cell lines stained with CellTrace^TM^Violet (Thermo Fisher Scientific, Waltham, MA, USA) at effector/target ratios of 5:1 and 15:1 in triplicate. In parallel, target cells were incubated alone to measure the basal cell death. Cells were incubated in V-bottomed 96-well microplates in a total volume of 150 μL of complete medium for 4 h in a 5% CO_2_ atmosphere at 37 °C. Cell mixtures were then washed in PBS-1% BSA and incubated in the same buffer containing 20 µg/mL 7-amino actinomycin D (7-AAD, Sigma-Aldrich, St. Louis, MO, USA) and 0.5 µg/mL DRAQ5 (BD, Mississauga, ON, Canada) for 10 min at 4 °C in the dark. Cells were then washed and acquired right afterwards for analysis in a FACSLyric flow cytometer. The mean value of triplicates was used to calculate the percentage of lysis as follows: experimental—spontaneous apoptotic target cells. The gating strategy is shown in Figure 1B.

### 2.5. Statistical Analysis

Data were collected in Excel 2010 (Microsoft Corporation, Seattle, WA, USA) and analyzed using SPSS-21.0 (SPSS, Chicago, IL, USA). Chi-square analysis and the analysis of variance/post hoc tests were used to analyze categorical and continuous variables, respectively. Kaplan–Meier and log-rank tests were used to analyze patient survival (PFS and OS). The time to events (progression or death) was estimated as months from the diagnosis date. The outcomes of patient groups, expressed in months, were estimated as the 75th-percentile-PFS (75p-PFS) or 75th-percentile-OS (75p-OS). Cox regression was used to investigate the effect of multiple parameters on the OS. The hazard ratio (HR) and 95% confidence interval were estimated. *p* < 0.05 was considered statistically significant. A Bonferroni correction (p_c_) was applied when necessary.

## 3. Results

### 3.1. Clinical, Biological, Therapeutic and Evolutionary Characteristics of the Study Groups

This study included 325 patients with BC, 150 patients with myeloma, 88 patients with ovarian cancer, 308 patients with melanoma, and 105 patients with pediatric acute leukemia with 75p-OS values of 34.04 ± 4.4, 35.0 ± 5.5, and 18.0 ± 2.9 years, respectively, and not reached for melanoma and pediatric acute leukemia (Figure 2). For BC, 57 were NIUN (CIS or Ta), and the rest UIC, with 148 T1, 99 T2, and 88 T3 or T4, and 75p-OS values of 120 ± 27.2, 53.0 ± 7.0, 14.0 ± 3.6, and 14.0 ± 2.9 years, respectively. BCG therapy was administered to 32 CIS + Ta cases (56.1%), 117 T1 cases (78.5%), and 2 T2 cases (2.0%). The 75p-OS values for BCG and other therapies were 59.0 ± 6.5 and 19.0 ± 2.9 years, respectively. The age and sex of the study groups are shown in Figure 2. The mean age of the HC group (n = 925) was 52 ± 0.7 years, with a proportion of 44.5% males.

### 3.2. KIR2DL3/C1 Was the Only Interaction Associated with Susceptibility to BC and Patient Outcome

We first assessed the role of NK cell receptor/ligand interactions in susceptibility to BC (Figure 3). The KIR3DS1 gene, but not the KIR3DS1/Bw4 interaction, showed a higher frequency in BC patients treated with BCG or other treatments (50.7% and 47.5%, *p* = 0.044) than in healthy controls (41.3%) or other cancers (40.3%); therefore, KIR3DS1 was associated with higher susceptibility to BC, although this molecule did not significantly impact disease progression or patient outcomes.

HLA C1 ligands showed a lower frequency in BC patients treated with BCG (70.0%, *p* = 0.038) than in BC patients undergoing other treatments (81.5%), in other cancers (80.0%), or in healthy controls (84.4%); therefore, C1 ligands were associated with lower susceptibility to low stages of BC, although this molecule did not significantly impact disease progression or patient outcomes.

KIR2DL3/C1 was the only interaction that a showed lower frequency in BC patients treated with BCG (61.8%, *p* < 0.001) than in healthy controls (75.8%), in BC patients treated with other therapies (68.9%), and in other cancer patients (70.2%); therefore, the KIR2DL3/C1 interaction appeared to protect against low-stage BC. Besides, although BC patients with the KIR2DL3/C1 interaction treated with BCG did not show differences in the progression-free survival curves, they showed longer 75p-OS times than those without this interaction (71.0 ± 12.0 vs. 56.0 ± 11.0 months, *p* = 0.031).

### 3.3. HLA-B −21M/T Genotype Is an Independent Predictive Parameter of the Progression-Free and Overall Survival in BC Patients Treated with BCG

Next, the role of the HLA-B −21M/T genotype in the recurrence and the progression of the disease and the survival of cancer patients was evaluated. In patients treated with BCG, disease recurrence was lower for the MM (29.4%) and MT (31.3%) genotypes than for the TT (37.3%) genotype, while in those patients treated with other therapies, only the MM genotype (18.8%) was associated with lower recurrence than MT (27.5%) and TT (24.7%). Nonetheless, none of these results reached the level of statistical significance.

In relation to the PFS and the OS, the −21M/T genotype showed opposite effects in BC patients treated with BCG and with other therapies (Figure 4A). In patients with other treatments, the MM genotype was associated with shorter 75p-PFS times (7.0 ± 2.2 vs. 16.0 ± 4.8 and 19.0 ± 4.7 months, *p* = 0.415) and 75p-OS times (8.0 ± 2.4 vs. 21.0 ± 3.4 and 19.0 ± 4.9 months, *p* = 0.131) than the MT and TT genotypes. However, in BCG-treated patients, the MM genotype was associated with longer 75p-PFS times (not reached vs. 47.00 ± 18.6 and 36.0 ± 12.0 months, *p* = 0.100) and 75p-OS times (not reached vs. 68.0 ± 13.7 and 52.0 ± 8.3 months, *p* = 0.034) than the MT and TT genotypes. In other cancers, the HLA-B −21M/T genotype was not associated with differential PFS or OS.

HLA −21M/T genotype was an independent predictive parameter of the PFS (HR = 2.08, *p* = 0.01) and the OS (HR = 2.059, *p* = 0.039) of BC patients treated with BCG, together with tumor size (HR = 3.064, *p* = 0.001), tumor pattern (HR = 2.224, *p* = 0.035), and tumor recurrence (HR = 3.198, *p* = 0.001) for PFS, and age (HR = 1.068, *p* = 0.007) and tumor staging (HR = 2.989, *p* = 0.064) for OS.

### 3.4. HLA-B −21M/T Genotype Is Associated with Differential Repertoire of KIR^+^ NK Cells and Expression of NKG2A in CD56^bright^ NK Cells

To evaluate the imprint that the HLA-B −21M/T genotype might have on antitumor effectors, the repertoire of NK and T lymphocytes and the expression of activating (CD226 and CD16) and inhibitory (TIGIT and NKG2A) receptors in NK cells were evaluated in the peripheral blood of 268 BC patients at diagnosis (Figure 5). No differences in the numbers of CD4^+^ and CD8^+^ T lymphocytes or CD56^dim^ NK cells were associated with the HLA-B −21M/T genotype. However, the MM genotype showed a higher frequency of CD56^bright^ NK cells (5.82 ± 1.07%, 4.12 ± 0.28% and 4.09 ± 0.28%, *p* < 0.05) and a lower frequency of NK cells expressing single-KIR2DL1^+^ (4.12 ± 0.58%, 6.54 ± 085% and 8.51 ± 0.83%, *p* < 0.05) and single-KIR2DL2/S2^+^ (6.07 ± 1.60%, 10.95 ± 1.47% and 10.14 ± 1.01%, *p* < 0.05) receptors than the MT and TT genotypes.

Although no differences in the expression of CD226, CD16, or TIGIT were associated with the HLA-B −21M/T genotype in CD56^dim^ and CD56^bright^ NK cells, the expression of NKG2A decreased proportionally to the number of −21M ligands (3.5 ± 0.50, 4.6 ± 0.45, and 5.9 ± 0.8 MFI × 10^3^, *p* < 0.05, for MM, MT, and TT genotypes, respectively) in CD56^bright^ cells but not in CD56^dim^ NK cells.

### 3.5. HLA-B −21M/T Genotype Was Not Associated with Differential NK Cell Functionality In Vitro

Finally, T and NK cell effector functions were evaluated in the PBMCs of 18 donors stimulated in vitro with anti-CD3/CD28 or BCG (Figure 6). Anti-CD3/CD28 mainly stimulated the proliferation of CD4+ T cells, whilst BCG strongly stimulated the proliferation of NK cells and the secretion of IL-1β, IL-6, IFN-γ, TNFα, and TGFβ1. Nonetheless, the HLA-B −21M/T genotype was not associated with any significant variations in the effector function of NK or T cells after anti-CD3/CD28 or BCG stimulation, in the secretion of cytokines, in the proliferation of T or NK cells, or in the cytotoxic capacity of NK cells.

## 4. Discussion

Following BCG treatment of NMIBC, a 32.6% to 42.1% local recurrence rate and a 9.5% to 13.4% progression rate are usually observed [53]. Unfortunately, radical cystectomy is the only definitive treatment option for BCG unresponsive disease, leaving patients at high risk of complications and a diminished quality of life [54]. Several immune-escape mechanisms associated with BCG therapy have been described, including the loss of HLA-I or increased PD-L1 expression in tumor cells [14,16]. Immunotherapeutic strategies, especially those engaging the PD-1/PD-L1 axis in order to unleash exhausted immune cells, have modestly improved BC outcomes [55]. The NKG2A/HLA-E interaction has been identified as a potent immune checkpoint regulating both CD8T and NK cells [56]. In fact, combined PD-1/PD-L1 and NKG2A/HLA-E immunotherapy has demonstrated an improved objective response in a Phase 2 clinical trial of patients with nonresectable, stage III non-small-cell lung cancer [57]. Recent results have shown that IFN-γ induced by BCG treatment enhances HLA-E and PD-L1 expression in recurrent tumors and NKG2A expression in intra-tumoral NK and CD8T cells and have provided a framework for a combined NKG2A and PD-L1 blockade strategy for the bladder-sparing treatment of BCG-unresponsive BC [12]. Although these strategies have not yet been adequately tested in BC, these results underscore the importance of combined strategies that enhance the synergy between NKG2A/HLA-E axis inhibition and other immune checkpoints in BC. The implementation of well-designed clinical trials in BC could validate the impact of combining NKG2A inhibitors with BCG or alternative therapies on the improvement of patient outcomes. Nonetheless, our results show that the role of the NKG2A/HLA-E interaction in the success of BCG therapy, and other therapies, might differ dramatically and might be linked to the HLA-B −21M/T genotype, with MM patients having very favorable outcomes if treated with BCG, but very unfavorable outcomes when treated with other therapies. These results appear to indicate that HLA-B genotyping could help to personalize these immunotherapies to improve the outcomes in BC.

In fact, the results presented in this work have two potential therapeutic implications. On the one hand, in NMIBC, BCG should be the treatment of choice in patients with the MM genotype, while the blockade of the NKG2A/HLA-E interaction should be questioned, since it could worsen the good results observed in these patients. On the other hand, blocking this immune checkpoint in patients who are going to receive non-BCG therapies would be fully justified; however, in view of the good results obtained with BCG therapy in patients with NMIBC, the use of BCG therapy should also be considered in MM patients with MIBC before radical cystectomy. Nonetheless, this decision should be based on new studies with larger series and properly randomized trials.

Therefore, the NKG2A/HLA-E licensing interaction appears to have a counteracting effect depending on the treatment. In patients with MIBC under conventional therapy, the inhibitory function of NKG2A in the presence of its specific ligand, HLA-E highly expressed by the double dose of −21M, seems to predominate [12,47,48]. These results are in agreement with evidence suggesting that −21M is unfavorable in the context of HIV-1 infection, as it is associated with the accelerated acquisition of HIV-1 infections [44] and lower NK cell cytotoxic capacity [45]. In contrast, the microenvironment induced by BCG stimulation seems to enhance the antitumor activity of NK cells that might have received a more potent education in the presence of −21M [31]. Again, this is in line with results describing higher leukemia-free survival and OS in −21M patients than in −21T AML patients during IL-2–based immunotherapy [46].

To try to understand the mechanism involved in this divergence, we analyzed the peripheral blood NK cell repertoire and the expression of activating and inhibitory receptors in BC patients at diagnosis. It was observed that MM genotypes were associated with higher frequencies of CD56^bright^ NK cells, which also showed lower NKG2A expression, indicating a strong interaction with the ligand HLA-E [33]. In addition, these patients had fewer single-positive KIR2DL1 and KIR2DL2 NK cells, although this may be due to the worse Bw4/C2 ligand-mediated licensing typically observed in MM genotypes [31,43]. However, the combination of lower NKG2A expression and lower levels of KIR2DL1+ NK cells could resemble combination therapy with Monalizumab (anti-NKG2A) and Lirilumab (anti-KIR2DL1/3). The combination of these therapies prevents tumor metastasis by activating the NK-mediated killing of circulating tumor cells [58]. However, these −21M/T genotype-induced differential features did not appear to be translated into the functionality of the NK cells when stimulated in vitro with BCG or into that of T cells stimulated with anti-CD3/CD28. Therefore, further multi-omics studies will be necessary to identify the mechanisms involved in this differential effect of −21M/T genotypes, mechanisms that could be of particular importance in BC as they do not seem to be active in other types of cancer.

Our study had clear limitations. On the one hand, the low frequency of the MM genotype (≈11.0% in BC patients and healthy controls) limited the follow-up to 17 MM cases in each of the treatment groups (BCG and other therapies); and on the other hand, the fact that we were not able to identify functional aspects in NK cells or T lymphocytes that could justify the differences in patient outcomes associated with the −21M/T genotypes, especially after identifying differences in their circulating NK cells. It is likely that our in vitro model was not able to emulate the complexity of the tumor microenvironment, and therefore, in order to corroborate these results, it will be necessary to perform in vitro 3D culture experiments to model the tumor microenvironment [59] and/or to develop experimental animal models. These studies should evaluate how the NKG2A/HLA-E interaction may influence the functionality of different immune effectors, such as NK and T lymphocytes [50], and its impact on the interaction of cancer cells with the stroma, as well as on the progression and the metastatic dissemination of the disease.

In conclusion, although our data should be confirmed in experimental models and clinical trials, they suggest that the study of the HLA-B −21M/T genotype could contribute to optimizing immunotherapy in patients with bladder cancer.

## Figures and Tables

**Figure 1 biomedicines-13-00156-f001:**
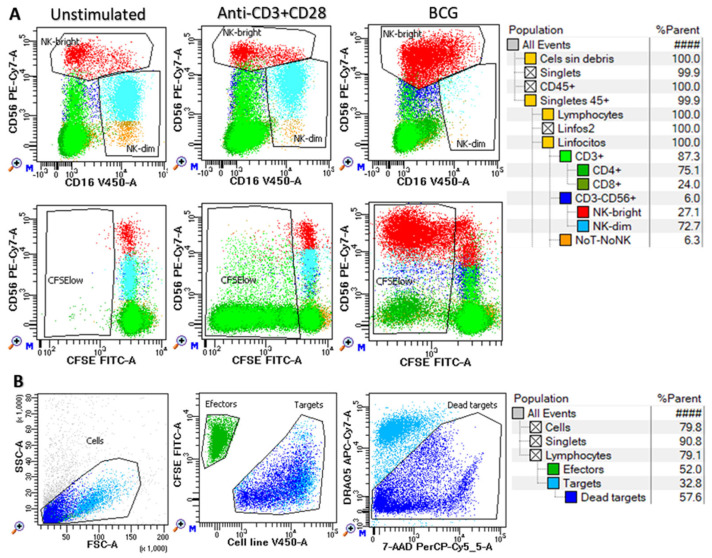
Functional assays of NK and T cells. Peripheral blood mononuclear cells were cultured in vitro with no-stimulation (unstimulated) or stimulated with anti-CD3/CD28 or BCG during 144 h to evaluate the following: (**A**) cell proliferation (CFSE-low cells) in CD3 + CD4 + (dark green) and CD3 + CD8 + (pale green) T lymphocytes as well as in CD56^dim^ (blue) and CD56^high^ (red) NK cells and noT-noNK cell subsets (orange); and (**B**) cytotoxicity against K562, T24 and J82 cell lines at different effector (green) to target (pale blue: alive and dark blue: dead) ratios. In both cases, a hierarchical and logical gating strategy was used in order to identify the corresponding cell subsets.

**Figure 2 biomedicines-13-00156-f002:**
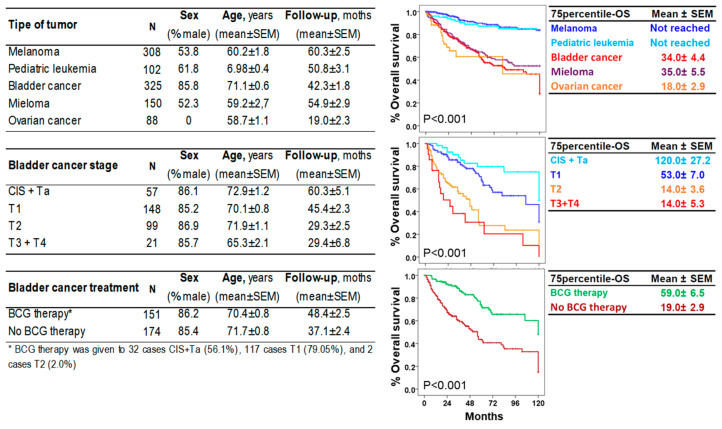
Clinical, biological, therapeutic and evolutionary (Kaplan–Meier and Log rank tests) characteristics of the study groups.

**Figure 3 biomedicines-13-00156-f003:**
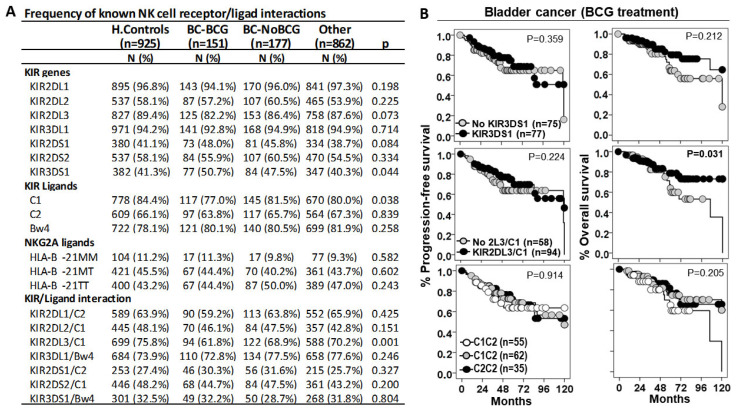
KIR2DL3/C1 is the only interaction associated with susceptibility to BC and patient outcome. (**A**) The frequency of known NK cell receptor/ligand interactions in the study groups. (**B**) Kaplan–Meier and log-rank tests for progression-free survival and overall survival according to the presence of KIR3DS1 gene, KIR2DL3/C1 interactions, and HLA C1 ligands.

**Figure 4 biomedicines-13-00156-f004:**
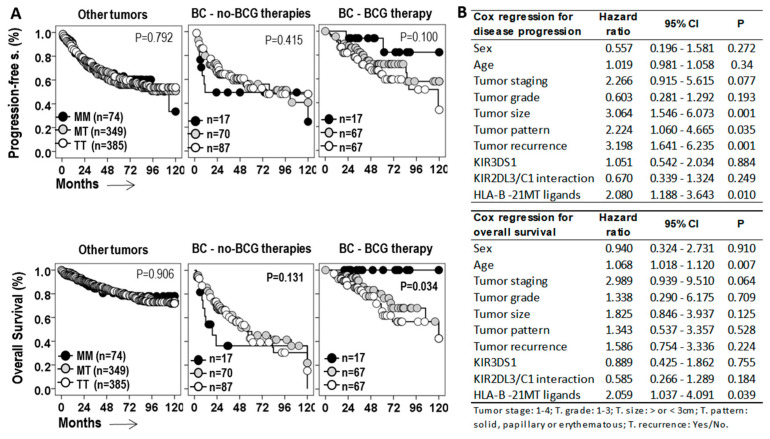
**The** HLA-B −21M/T genotype is an independent predictive parameter of the progression-free and overall survival of BC patients treated with BCG. (**A**) Kaplan–Meier and log-rank test for progression-free survival (PFS) and overall survival (OS) of patients with tumors other than bladder cancer (BC) and for patients with BC treated with BCG or other therapies according to the HLA-B −21M/T ligand genotype. (**B**) Cox regression analysis for PFS and OS of BC patients treated with BCG for age, tumor staging, grade, size, pattern, and recurrence, and in terms of HLA-B −21M/T ligand genotype.

**Figure 5 biomedicines-13-00156-f005:**
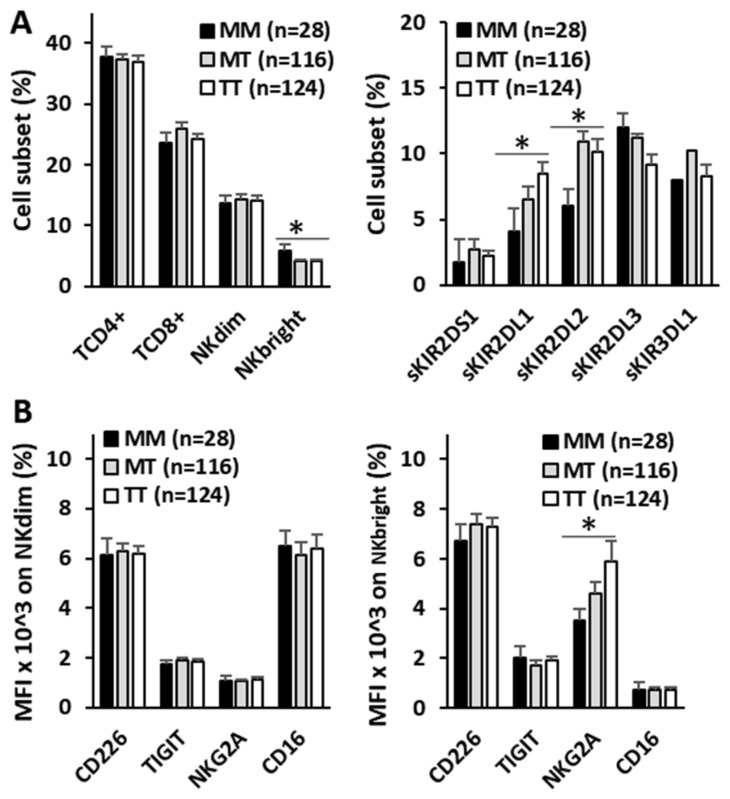
The repertoire of T lymphocytes and NK cells in peripheral blood and the expression of NK cell-activating and -inhibitory receptors of bladder cancer patients. A) The frequency of CD4+ and CD8+ T lymphocytes and CD56^dim^ and CD56^bright^ NK cells and the NK single-KIR^+^ (sKIR) repertoire according to the HLA-B −21M/T ligand genotype. B) Mean fluorescence intensity (MFI) expression of activating (CD226 and CD16) and inhibitory (TIGIT and NKG2A) receptors on CD56^dim^ and CD56^bright^ NK cells, according to the HLA-B −21M/T ligand genotype. *, *p* < 0.05 in the ANOVA test.

**Figure 6 biomedicines-13-00156-f006:**
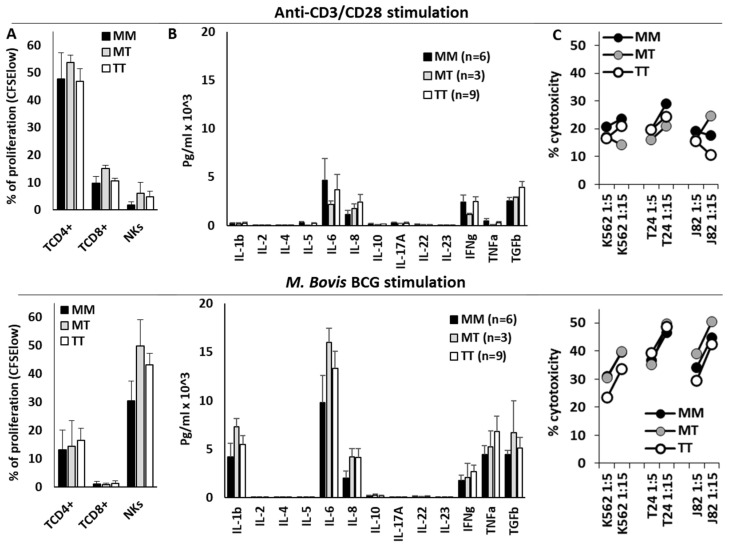
The HLA-B −21M/T genotype was not associated with differential NK cell functionality after anti-CD3/CD28 or BCG in vitro stimulation. The proliferation (% of CFSE-low cells) of CD4+ and CD8+ T lymphocytes and CD56 + CD3- NK cells (**A**), cytokine secretion to the supernatant (**B**), and cytotoxicity against K562, T24, and J82 cell lines (**C**) of PBMCs stimulated with anti-CD3/CD28 (upper plots) or BCG (lower plots) according to the HLA-B −21M/T genotype.

## Data Availability

The data of the work will be available upon justified request of the researchers who need it.

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
