# Peer review of "Differential Role of NKG2A/HLA-E Interaction in the Outcomes of Bladder Cancer Patients Treated with *M. bovis* BCG or Other Therapies"

_biomedicines, 2025, doi:10.3390/biomedicines13010156_

Round 1
Reviewer 1 Report
Comments and Suggestions for Authors
1. Abstract-does not reflect the wide population enrolled beyond bladder cancer. Please clarify.
2. Endpoint- overall survival is not optimal as an endpoint to study NMIBC since this stage does not cause much mortality.
3. Methods- the choice of endpoint needs explanation. Survival in not ideal for non-invasive cancers. Recurrence and progression are better endpoints for non-invasive locally recurrent tumors.
4. The relevance of healthy controls and non-bladder cancer cases is unclear. Please justify and discuss and mention in abstract too.
5. Were analyses controlled for stage, grade and therapy?
6. Discussion could highlight some emerging data from clinical trials targeting the NKG2A axis.
Comments on the Quality of English LanguageReasonable
Author Response
Comments 1: Abstract-does not reflect the wide population enrolled beyond bladder cancer. Please clarify.
Response 1: We have added other cancers and health donors as control groups.
Comments 2: Endpoint- overall survival is not optimal as an endpoint to study NMIBC since this stage does not cause much mortality.
Response 2: It is true that for this stages the endpoints in clinical trials are usually based on disease recurrence and progression, mainly because they are faster endpoints to reach and therefore allow earlier evaluation. However, as can be seen in our work, not only disease progression is affected by NKG2A/HLA-E interaction but also patient survival. In fact, this second parameter evaluates BETTER the impact of the patient's immunogenetics than progression and relapse, in which perhaps the genetic components of the cancer itself have greater weight. When assessing survival, the results indicate how the patient's immunity manages with the progression and expansion of the tumor that will finally cause death. In any case, in the section 3.3., the impact of the M/T polymorphism on tumor recurrence has been described in the manuscript; As it can be seen, shows a trend to a lower recurrence rate in M genotypes, although these results do not reach statistical significance either in patients treated with BCG or in those treated with other therapies.
Comments 3: Methods- the choice of endpoint needs explanation. Survival in not ideal for non-invasive cancers. Recurrence and progression are better endpoints for non-invasive locally recurrent tumors.
Response 3: Following the argumentation of the previous point, at the end of the first paragraph of section 2.1 it has been added that “Overall survival of patients, in addition to disease recurrence and progression, was in-cluded as an endpoint in both treatments, since the aim of our study was to evaluate the role of NKG2A/HLA-E interaction in local and systemic tumor immune response”.
Comments 4: The relevance of healthy controls and non-bladder cancer cases is unclear. Please justify and discuss and mention in abstract too.
Response 4: We have modified the beginning of the 2.1 section to say “This prospective observational study included 325 consecutive patients diagnosed with BC. As control groups, 925 healthy Caucasian patients (HC) and 973 patients with other types of tumor (308 melanomas, 150 myelomas, 102 pediatric acute leukemias, and 88 ovarian cancers) were included for the purpose of identifying immunological parameters that specifically influence susceptibility to BC and response to different therapies, including BCG immunotherapy”. We have also introduced briefly in the abstract the porpoise of the inclusion of these groups.
Comments 5: Were analyses controlled for stage, grade and therapy?
Response 5: As stated in the manuscript (Figure 2), basically stage and grade conditioned treatment with BCG or other therapies, so indirectly the analyses were controlled for these parameters. However, to evidence the independent role of NKG2A/HLA-E interaction in disease progression and patient survival, in Figure 4 the variables of stage, grade, size, pattern and recurrence were included in the linear and cox regression analyses. So that all the histological parameters were evaluated together with the NKG2A/HLA-E interaction.
Comments 6: Discussion could highlight some emerging data from clinical trials targeting the NKG2A axis.
Response 6: To our knowledge, there are no clinical trials in bladder cancer that would allow us to discuss the relevance of our findings for the future development of new immunotherapies. However, in the first paragraph of the discussion we have discussed the relevance that the NKG2A/HLA-E interaction could have in the design of future trials to evaluate the combination of different immune checkpoints in patients treated with BCG or other therapies.
Reviewer 2 Report
Comments and Suggestions for Authors
The authors should talk in more detail in the introduction about immunotherapy and the escape mechanism that occurs in bladder cancer at this reference doi: 10.3390/ijms25126750. is useful. Perform additional in vitro and in vivo studies to confirm the findings and investigate the mechanisms underlying the differential effect of the HLA-B-21M/T genotype. The authors should explore the potential impact of other immune checkpoint interactions on BC outcomes. Discuss the limitations of the study in more detail. The in vitro functional assays may not accurately reflect the complexity of the tumor microenvironment, and further studies using 3D cultures or animal models are needed to confirm the findings. The authors do not provide a clear explanation for the differential effect of the HLA-B-21M/T genotype depending on the treatment received.
Comments on the Quality of English Language
The authors should improve the fluency and pay attention to some grammatical errors.
Author Response
Comments 1: The authors should talk in more detail in the introduction about immunotherapy and the escape mechanism that occurs in bladder cancer at this reference doi: 10.3390/ijms25126750. is useful.
Response 1: The role of immunotherapy with anti- checkpoint has been expanded and the escape mechanisms occurring in BC, including those in BCG treatment and those described in the suggested citation, have been described.
Comments 2: Perform additional in vitro and in vivo studies to confirm the findings and investigate the mechanisms underlying the differential effect of the HLA-B-21M/T genotype.
Response 2: As described in the discussion of our work, we propose future studies that address new in vitro models (three-dimensional models) to better mimic the complex tumor microenvironment and in vivo models to discern the relevance of the different components of the immune system, including CD8 T lymphocytes. However, to carry out these studies it will be necessary to obtain new funds. The publication of this work will contribute very significantly to the possibilities of obtaining funding and being able to extend this study as you propose.
Comments 3: The authors should explore the potential impact of other immune checkpoint interactions on BC outcomes.
Response 3: Our study was focused on evaluating the role of NK cells in bladder cancer immune response, and immune checkpoints such as CTLA-4 or PD-1/PD-L1 have not been studied. However, we have explored the expression of TIGIT on NK and T cells and we could not find any relation with recurrence, progression or survival of patients. As can be seen in figure-4. The impact of the interactions of KIR2DL5 and other KIRs with their ligands as a function of BCG treatment is currently being evaluated. But these results will be the subject of further work.
Comments 4: Discuss the limitations of the study in more detail. The in vitro functional assays may not accurately reflect the complexity of the tumor microenvironment, and further studies using 3D cultures or animal models are needed to confirm the findings.
Response 4: We have expanded the discussion of this limitation.
Comments 5: The authors do not provide a clear explanation for the differential effect of the HLA-B-21M/T genotype depending on the treatment received.
Response 5: We would like nothing more than to be able to give a reasonable explanation for the differences in survival observed for the -21 M/T dimorphism in patients treated or not with BCG. There are no precedents in the literature to provide such an explanation. However, the discussion has been extended to discuss this point in relation to the few studies describing the role of this dimorphism in HIV infection and in IL2 immunotherapy in AML. Indeed, understanding these differences could allow a more personalized treatment strategy and to improve outcomes in bladder cancer.
Reviewer 3 Report
Comments and Suggestions for Authors
The article titled “Differential role of NKG2A/HLA-E interaction in the outcome of bladder cancer patients treated with M. bovis BCG or other therapies” discussed the relationship between HLA-B -21M/T genotype and bladder cancer patients treated with BCG, this article has sufficient d data, but some concerns need to be addressed:
1. The authors should introduce MM, MT and TT genotype and differences among them in the introduction section.
2. In figure 1, the total N number of 4 bladder cancer staging is 326, this is not consistent with the total N number (325) in this study, please check.
3. Since the authors concluded that HLA-B -21M/T genotype is independently associated with BC patient outcome, how can this conclusion contribute to optimize immunotherapy in patients with bladder cancer? The authors should discuss this issue.
4. In figure 1, corresponding statistical histogram figure should be added.
5. In line 70, the citation format and order of references is not correct.
6. Moderate professional English editing should be performed.
Comments on the Quality of English LanguageModerate professional English editing should be performed.
Author Response
Comments 1: The authors should introduce MM, MT and TT genotype and differences among them in the introduction section.
Response 1: We sincerely believe that the role of -21 M/T dimorphism in the immune response has been clearly and sufficiently described in the introduction, lines 94-108 from the new manuscript.
Comments 2: In figure 1, the total N number of 4 bladder cancer staging is 326, this is not consistent with the total N number (325) in this study, please check.
Response 2: Sorry, numbers have been fixed. Thanks.
Comments 3: Since the authors concluded that HLA-B -21M/T genotype is independently associated with BC patient outcome, how can this conclusion contribute to optimize immunotherapy in patients with bladder cancer? The authors should discuss this issue.
Response 3: This aspect is the main contribution of our manuscript, and although it was already discussed, we have tried to further analyze these results in the context of current immunotherapy trials ongoing in bladder cancer, in lines 373-383, 384-392, and 393-403.
Comments 4: In figure 1, corresponding statistical histogram figure should be added.
Response 4: We have added the statistical data to the legends.
Comments 5 :In line 70, the citation format and order of references is not correct.
Response 5: Fixed.
Comments 6: Moderate professional English editing should be performed.
Response 6: The manuscript was professionally reviewed, but we have sent the manuscript back for a new English revision. Hope now is better. Changes has been highlighted in yellow.
Round 2
Reviewer 3 Report
Comments and Suggestions for Authors
This manuscript has been sufficiently improved to warrant publication in Biomedicines.